# Phenotyping the virulence of SARS-CoV-2 variants in hamsters by digital pathology and machine learning

**Gavin R. Meehan**[1,2⦿]**, Vanessa Herder**[1,2⦿]**, Jay Allan**[1]**, Xinyi Huang**[1]**, Karen Kerr**[1,2]**, Diogo Correa Mendonca**[1,2]**, Georgios Ilia**[1]**, Derek W. Wright**[1]**, Kyriaki Nomikou**[1†]**, Quan Gu**[1]**, Sergi Molina Arias**[1,2]**, Florian Hansmann**[3]**, Alexandros Hardas**[4]**, Charalampos Attipa**[5]**, Giuditta De Lorenzo**[1]**, Vanessa Cowton**[1]**, Nicole Upfold**[1,2]**, Natasha Palmalux**[1]**, Jonathan C. Brown**[6]**, Wendy S. Barclay**[6]**, Ana Da Silva Filipe**[1]**, Wilhelm Furnon**[1]**, Arvind H. Patel**[1,2]\*, **Massimo Palmarini**[1]\*

**1** MRC-University of Glasgow Centre for Virus Research, United Kingdom, **2** CVR-CRUSH, MRC-University of Glasgow Centre for Virus Research, United Kingdom, **3** Institute of Veterinary Pathology, Faculty of Veterinary Medicine, Leipzig University, Germany, **4** Department of Pathobiology & Population Sciences, The Royal Veterinary College, North Mymms, United Kingdom, **5** The Royal (Dick) School of Veterinary Studies, The University of Edinburgh, United Kingdom, **6** Department of Infectious Disease, Imperial College London, United Kingdom

⦿ These authors contributed equally to this work.
† Deceased.
\* arvind.patel@glasgow.ac.uk (AHP); massimo.palmarini@glasgow.ac.uk (MP)

**Data Availability Statement:** The raw FASTQ files generated during this project have been submitted to the European Nucleotide Archive (ENA) under project accession number PRJEB55782. Raw data

## Abstract

Severe acute respiratory syndrome coronavirus 2 (SARS-CoV-2) has continued to evolve throughout the coronavirus disease-19 (COVID-19) pandemic, giving rise to multiple variants of concern (VOCs) with different biological properties. As the pandemic progresses, it will be essential to test in near real time the potential of any new emerging variant to cause severe disease. BA.1 (Omicron) was shown to be attenuated compared to the previous VOCs like Delta, but it is possible that newly emerging variants may regain a virulent phenotype. Hamsters have been proven to be an exceedingly good model for SARS-CoV-2 pathogenesis. Here, we aimed to develop robust quantitative pipelines to assess the virulence of SARS-CoV-2 variants in hamsters. We used various approaches including RNAseq, RNA *in situ* hybridization, immunohistochemistry, and digital pathology, including software assisted whole section imaging and downstream automatic analyses enhanced by machine learning, to develop methods to assess and quantify virus-induced pulmonary lesions in an unbiased manner. Initially, we used Delta and Omicron to develop our experimental pipelines. We then assessed the virulence of recent Omicron sub-lineages including BA.5, XBB, BQ.1.18, BA.2, BA.2.75 and EG.5.1. We show that in experimentally infected hamsters, accurate quantification of alveolar epithelial hyperplasia and macrophage infiltrates represent robust markers for assessing the extent of virus-induced pulmonary pathology, and hence virus virulence. In addition, using these pipelines, we could reveal how some Omicron sub-lineages (e.g., BA.2.75 and EG.5.1) have regained virulence compared to the original BA.1. Finally, to maximise the utility of the digital pathology pipelines reported in our study, we developed an online repository containing representative whole organ histopathology sections that can

underpinning the graphs presented in this study, and the training files necessary to implement the machine learning-assisted digital pathology pipeline necessary to detect alveolar epithelial hyperproliferation in HALO are available in the Elighten repository (https://doi.org/10.5525/gla.researchdata.1513).

**Funding:** Funding was provided by the the Wellcome Trust (206369/Z/17/Z to MP), the UKRI UK-G2P consortium (MR/W005611/1 to MP, AHP and WSB), by LifeArc (COVID-19 grant to MP and AHP) and MRC (MC MC_UU_00034/5 to QG and DWW and MC_UU_00034/9 to MP and AHP). The funders had no role in study design, data collection and analysis, decision to publish, or preparation of the manuscript.

**Competing interests:** The authors have declared that no competing interests exist.

be visualised at variable magnifications (https://covid-atlas.cvr.gla.ac.uk). Overall, this pipeline can provide unbiased and invaluable data for rapidly assessing newly emerging variants and their potential to cause severe disease.

## Author summary

New SARS-CoV-2 variants have periodically emerged throughout the COVID-19 pandemic. The key characteristic possessed by "successful" variants in this phase of the pandemic is their ability to evade the immunity conferred by existing SARS-CoV-2 vaccines or infections. However, it is not clear whether new variants will maintain the relatively low virulence shown by Omicron or acquire the more virulent phenotype of the pre-omicron variants.

In this study, we developed software-assisted image analysis methods to quantitatively assess the extent of the lesions in lungs of hamsters experimentally infected with SARS-CoV-2. Hamsters are an excellent animal model to assess SARS-CoV-2 virulence. Based on previous experiments, we reasoned that the virulence of SARS-CoV-2 variants will be directly proportional to the lung lesions they cause in hamsters. We developed unbiased methods aimed to image whole lung sections, and quantify immune cells infiltrating the organ, in addition to the levels of lung cells proliferation in response to virus injury. Using these methods, we show that Omicron is, as expected, less virulent than the Delta variant. The virulence of other variants such as XBB and BQ.1.18 is comparable to that displayed by Omicron. However, the more recently emerged variants BA.2.75 and EG.5.1 are more virulent than Omicron, but not Delta. Our methods can provide a quantitative assessment of the ability of newly emerging variants to cause severe disease.

## Introduction

As the coronavirus disease-19 (COVID-19) pandemic progressed over the past three years, the virus responsible for the disease, severe acute respiratory syndrome coronavirus 2 (SARS-CoV-2), has continued to evolve giving rise to a number of variants, some of which were defined as "variants of concern" (VOCs) by the World Health Organisation (WHO) [1, 2]. These VOCs contain mutations, especially but not solely in the spike encoding S-gene, which may confer a selective advantage for example by increasing their transmissibility and/or immune evasion compared to the progenitor virus [1]. To date, the WHO has recognised five VOCs: B.1.1.7 (Alpha); B.1.351 (Beta); P.1 (Gamma); B.1.617.2 (Delta) and B.1.1.529 (Omicron; henceforth referred as BA.1 to differentiate it from other sub-lineages) [1, 3–5]. Since the emergence of the original BA.1 in November 2021 [5] different sub-lineages have emerged. Soon after the BA.1 emergence, BA.2 became the predominant variant followed by a variety of BA.2 descendants, including BA.5 and BA.2.75, which became predominant in some geographical regions [1, 6]. Although both BA.5 and BA.2.75 diversified from BA.2, these two Omicron sub-lineages are phylogenetically separated from each other, suggesting that BA.5 and BA.2.75 emerged independently. BQ.1.18 is a sub-lineage of BA.5 [7] while the XBB variant is a recombinant between BJ.1 (a BA.2.10 derivative) and BM.1.1.1 (a descendant of BA.2.75), which was first detected in September 2022 in India and spread significantly at the end of 2022 [8]. EG.5.1 is a descendent lineage of XBB.1.9.2 but carries two additional spike mutations F456L (defining EG.5 lineage) and Q52H. It was first detected in early March 2023

and was then designated as a variant of interest (VOI) by WHO on 9 August 2023 along with EG.5 and its sub-lineages. EG.5.1 has been detected globally with a rapid increased prevalence to around 50% as of 2*nd* October 2023 [9].

In general, each new variant spreading globally tends to be more transmissible than the previous dominant variant [1]. As population immunity increases, due to either vaccination or continuous virus exposure, it is likely that COVID-19 will adopt an endemic pattern, possibly with seasonal peaks, primed by variants evading pre-existing immunity in the population [10].

Understanding in "real-time" the degree of vaccine escape of any new variant is critical to determine vaccination policies. To this end, *in vitro* seroneutralisation assays have proven to be a useful surrogate to predict vaccine escape of SARS-CoV-2 variants. Virus virulence is another key phenotypic characteristic of any new variant requiring early assessment. The risk of severe disease and hospitalisation varies, with Alpha, Gamma and Delta VOCs carrying an increased risk of intensive care unit (ICU) admission compared to Beta and BA.1. Hence, the combination of increasing pre-existing immunity in the population together with the intrinsic attenuated characteristics of Omicron, has led to an overall decrease in the incidence of severe disease and mortality associated with COVID-19 [11–14].

It is however more difficult to predict the trajectory of new variants with respect to virus virulence. Although BA.1 has been shown to be attenuated, and assuming that its transmission potential is maintained, there are no universal evolutionary pressures that may keep this phenotypic trait in newly emerging sub-lineages or new VOCs.

We and others have shown that the observed reduction in virulence of the BA.1 variant correlates to a change in the virus entry pathways *in vitro*, and importantly in reduced virulence in experimentally infected hamsters [15–18]. Throughout the pandemic, small animal models have been used extensively to assess the virulence of wild type SARS-CoV-2 and emerging VOCs [8, 15, 17–27]. These studies have provided invaluable data on disease pathogenesis, virus transmission and the efficacy of different anti-viral compounds or vaccines [20, 28–33]. Importantly, hamsters have been shown to be naturally susceptible to SARS-CoV-2 infection and to be able to transmit the virus to humans [34]. In hamsters, BA.1 is unable to infect lung epithelial cells (unlike the original B.1 virus, Delta and other variants) [15, 17]. In addition, in experimentally infected hamsters it is also possible to recapitulate the increased virulence of the Delta variant compared to B.1 shown in the human population [24, 35].

Hence, although no animal model can fully recapitulate a human disease, hamsters represent an excellent model to dissect SARS-CoV-2 pathogenesis and determine the degree of virulence of newly emerging variants. To this end, many studies using experimentally infected hamsters, have focused on measuring *in vivo* viral replication, on identifying virus infected cells, and on examining pathogenic potential by measuring weight loss and assessing various histopathological criteria in general by qualitative scores [8, 18–21]. Here, we have developed unbiased and automated "digital pathology" methods to assess SARS-CoV-2 virulence. Digital pathology is a broad term that refers to a variety of systems to digitize pathology slides and associated meta-data, their storage, review, analysis, and enabling infrastructure [36]. Computational analysis of whole scanned tissue sections provides the opportunity to quantify cells or histological features in wide representative areas of infected organs. We applied these pipelines with recently evolved variants (BA.5, BQ.1.18, BXX, BA.2 and BA.2.75) [1, 6–8] and showed that some of them have gained a more virulent phenotype compared to the parent BA.1. This pipeline can contribute to the rapid assessment of newly emerging variants and should prove invaluable as the pandemic enters the next phase. Furthermore, we created an online repository to share with the scientific community high resolution digitized whole organ scanned slides from this study, providing a wider context to histopathology micrographs for experimental models of COVID-19.

## Results

### Host transcriptional response to SARS-CoV-2 infection

To assess the complex host responses during SARS-CoV-2 infection, we first experimentally infected Golden Syrian hamsters with either the Delta (B.1.617.2) or the BA.1 variants. These variants are on the opposite spectrum of the phenotype associated with the clinical outcome of SARS-CoV-2 infection, both in humans and in experimental models. Hence, Delta and BA.1 can provide the baseline for the development of quantitative pipelines to assess virus virulence. As expected, between 2- and 6-days post-infection (dpi) the Delta-infected animals lost significantly more weight and had higher welfare scores than both the BA.1- and mock-infected hamsters (S1A Fig), confirming the expected phenotype for both variants. We culled animals at either 2 or 6 dpi and collected tissues of both the upper and lower respiratory tract, in addition to peripheral blood, for downstream analyses.

To understand the overall host responses to SARS-CoV-2 infection and identify potential markers of virus virulence, we performed bulk RNAseq on lungs and blood of both infected and mock-infected hamsters. Principal component analysis (PCA) indicated distinct clustering of both the Delta- and BA.1-infected groups at 2 dpi in lungs and blood but showed less distinctive separation of BA.1 and mock-infected animals at 6 dpi (S1B Fig). Both in the lungs and in peripheral blood, Delta, and BA.1 induced significant differential gene expression compared to the mock-infected samples (Fig 1A). As expected, both variants induced multiple cell responses pathways associated with antiviral mechanisms, immune system activation, cytokine and chemokine responses and interferon signalling as evident by gene ontology (GO) analysis (Fig 1B and 1C). Direct GO analysis between the two variants found a limited number of pathways on day 6 upregulated only in delta-infected animals. These pathways include those associated with remodelling and lipoprotein regulation, suggesting that the differences in disease outcome caused by these two variants may be attributed to lesions and activation of tissue repair pathways in the lungs. Comparison of ~200 interferon stimulated genes showed a general activation of the type-I IFN response in infected animals (as suggested by the GO analysis and previously published studies) both in the lungs and in the blood [37–39]. Delta-infected animals showed a more robust upregulation of interferon stimulated genes (ISGs) than BA.1-infected hamsters in both lungs and blood (Fig 1D and 1E). Overall, these analyses suggest that markers of type-I IFN response, immune system activation and tissue repair may be useful to characterise the extent of pathology induced by SARS-CoV-2 variants.

### Imaging and quantification of SARS-CoV-2 replication in tissues

Next, we characterised the spread of SARS-CoV-2 infection in infected tissues. Throughout this study, for the detection of virus and cellular markers, we aimed to develop unbiased quantitative methods using software assisted whole section imaging and downstream automatic analyses including those enhanced by machine learning approaches (henceforth referred with the broad term of digital pathology) [40]. The development of the digital pathology pipeline presented in this study has been evaluated by a board-certified veterinary pathologist (VH). In addition, side-by-side comparisons between standard histopathological scores and the digital pathology pipeline was carried out by three additional board-certified veterinary pathologists (FH, AH, CA). Also, in order to share the histopathology features shown in this study as comprehensively as possible, we have developed an online repository ("CVR Virtual Microscopy"; https://covid-atlas.cvr.gla.ac.uk) where whole organ scanned images can be accessed by users in their entirety and at variable magnification as if they were observing slides under a microscope.

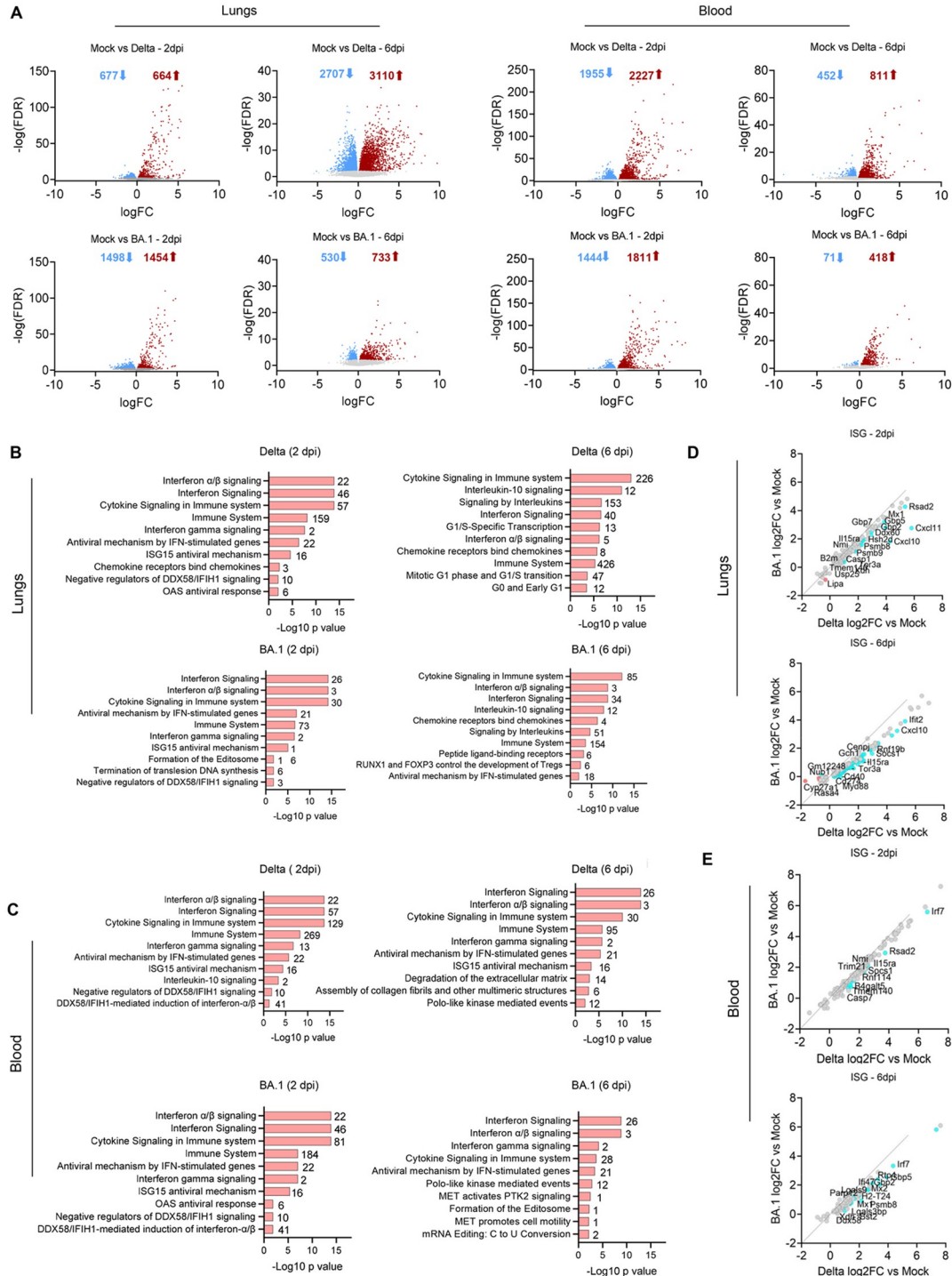

**Fig 1. Transcriptomic response of lungs and blood of SARS-CoV-2 infected hamsters.** (A) Volcano plots indicate the number of significantly upregulated (red arrows) or downregulated (blue arrows) genes in lungs or blood of hamsters infected with the indicated variant at each timepoint (compared to mock-infected controls). (B-C) Pathway analysis highlighting the enriched ontologies of differentially expressed genes in the lungs (B) and blood (C) of infected hamsters at 2—and 6-days post-infection (dpi). The number of genes involved are indicated beside each pathway. (D-E) Scatterplots representing the relative expression of interferon stimulated genes (ISG) in lungs (D) and blood (E) of hamsters experimentally infected with either the Delta or BA.1 variant at the indicated timepoints. ISG with relative higher expression in BA.1 compared to Delta (FDR<0.05) are shown in red. ISG with relative higher expression in Delta compared to BA.1 (FDR<0.05) are shown in blue. Data derived from n = 8 hamsters (4 females and 4 males per group), infected in two independent experiments.

First, we compared the detection levels of SARS-CoV-2 nucleocapsid protein in the lungs of infected hamsters at 2 dpi by immunohistochemistry (IHC) with its spike RNA by *in situ*-hybridisation. The two methods resulted in essentially identical staining patterns (S2 Fig). Background staining in mock-infected samples was higher using IHC and therefore we used RNA *in situ*-hybridisation for the remaining part of the study. Next, we assessed virus replication for Delta and BA.1 along the entire respiratory tract. As expected, tissues collected from animals culled at 2 dpi showed both Delta- and BA.1-infected cells within the respiratory tract (Fig 2A and 2B). We found instead little evidence of virus infected cells in animals culled at 6 dpi. At 2 dpi, we found no significant differences in the number of infected cells in the nose, larynx, and trachea of Delta- and BA.1-infected animals. In our samples, the nose represents the inner mucosa of the small cartilaginous tissue surrounding the nasal cavities of the hamster. However, both nasal turbinates of Delta-infected animals and the lungs showed a significantly higher number of infected cells than the same tissues in BA.1-infected hamsters. Importantly, significant spread of SARS-CoV-2 in the lung parenchyma was evident only in Delta-infected hamsters. Delta infected both epithelial cells in the bronchioles and alveoli forming large foci of infected tissues. Conversely, BA.1 infected only cells in the bronchioles and at most formed small foci of infection in the lung parenchyma in some animal (Fig 2A and 2B). As expected, there was variability between animals within each group with respect to the number of infected cells. This variability was especially evident in the trachea, with 2 of 8 Delta-infected animals showing a number of infected cells more than 10-fold the number in the remaining animals of the group. Except for the two outliers indicated above, the tracheas of the remaining Delta- and BA.1-infected hamsters showed a similar number of infected cells.

Given the data obtained by RNAseq, where the interferon response is a key differentially activated pathway in infected hamsters, we also assessed expression of MX1, as a representative core ISG [41]. MX1 expression in the nose and lungs was in general lower at 2 dpi compared to the levels observed at 6 dpi. The larynx showed high MX1 expression on 2 dpi but little on 6 dpi, while the turbinates and trachea showed similar levels at both timepoints (Fig 2C). Bulk RNAseq data suggested a more robust type-I IFN response elicited by the Delta variant compared to BA.1 in lungs from infected hamsters (Fig 1B–1E). We aimed therefore to spatially resolve ISG expression in infected hamsters using *in situ* RNA hybridisation of serial sections of lungs collected at 2 dpi. We used five sections with the middle section probed for spike, while in the other sections we used RNA probes for the following ISGs: RSAD2, IFIT1, MX1 and OAS1 (Fig 3). The percentage of ISG-positive pixels was directly related to the percentage of spike-positive pixels in the lungs of infected hamsters. There were clear overlapping virus- and ISG-positive areas in both Delta- and BA.1-infected lungs. Hence, the more robust type-I IFN response observed by RNAseq in the lungs and blood of infected animals correlates with higher replication levels of Delta in the lungs at 2 dpi.

The differences between Delta- and BA.1-infected hamsters were also present at earlier timepoints. Analysis of tissues collected at 1 dpi of the turbinates, trachea, and lungs (S3A Fig) showed increased levels of spike RNA (S3B Fig) in Delta-infected animals. MX1 expression showed a trend of higher levels of expression in Delta-infected animals but differences were not statistically significant (S3C Fig). Similarly, nasal washes, throat swabs and whole lung RT-qPCR indicated higher levels of genomic RNA in Delta-infected hamsters, but only in the latter differences were statistically significant (S3D Fig).

## Quantifying the extent of pulmonary lesions by digital pathology

We showed above that at 6 dpi, at the peak of clinical signs in experimentally infected hamsters, most of SARS-CoV-2 has been cleared by the host (Fig 2A). Our RNAseq analysis

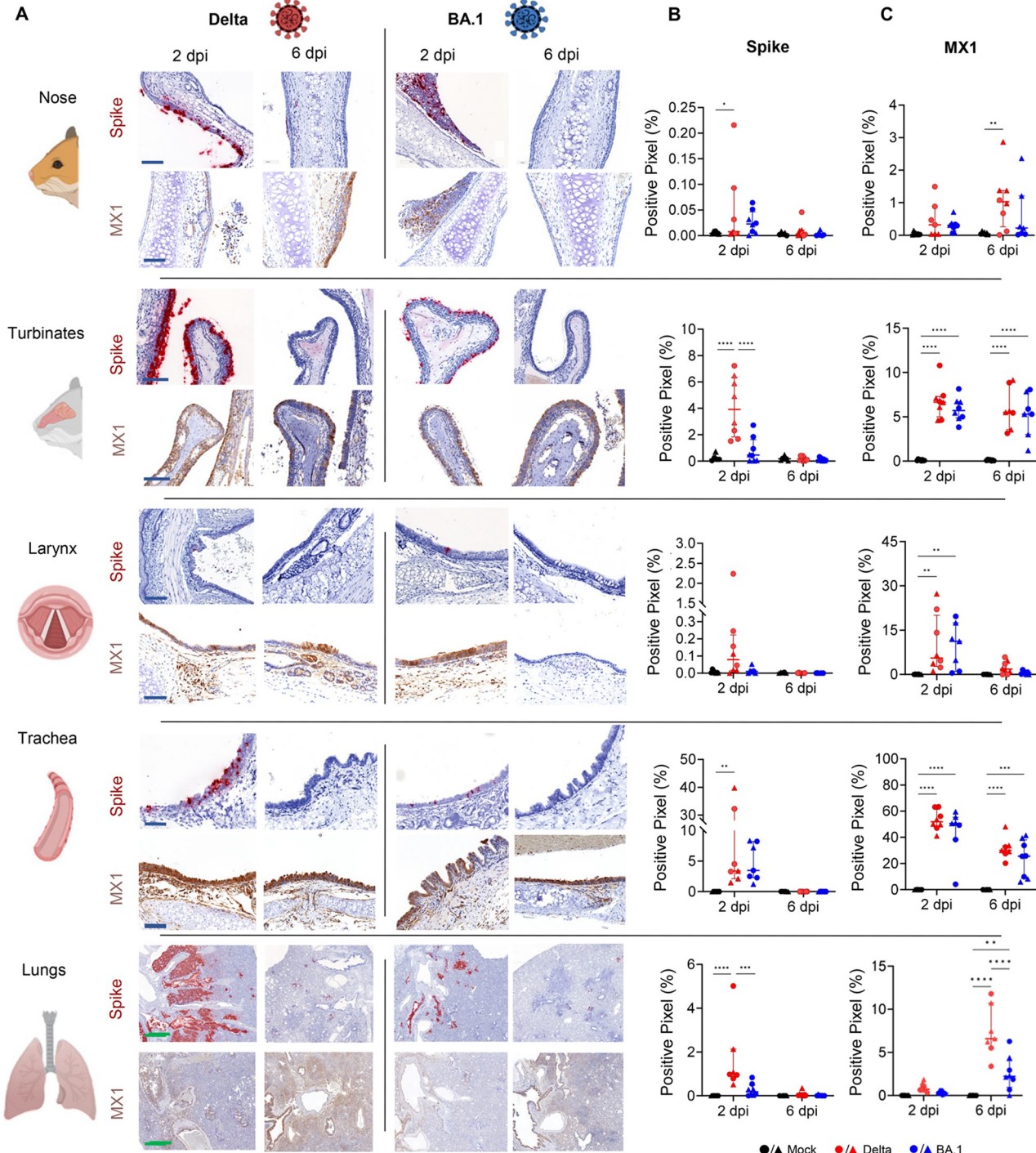

**Fig 2. Distribution of Delta and BA.1 in organs of the respiratory tract of experimentally infected hamsters.** Golden Syrian hamsters were infected intranasally with either Delta or BA.1 ($3.75 \times 10^6$ genome copies equivalent per animal) (or mock-infected). Control animals received media alone. Animals were culled 2- or 6-days post infection (dpi) and the nose, turbinates, larynx, trachea, and lungs were collected for digital pathology analyses (A). Tissues were assessed for the presence of spike RNA by *in situ* hybridisation (B) or for the expression of MX1 by immunohistochemistry (C). For signal quantification, slides were scanned with an Aperio VERSA 8 Brightfield, Fluorescence & FISH Digital Pathology Scanner (Leica Biosystems) at 200 x brightfield magnification. Tissues to be analysed were outlined using QuPath (Version 0.3.2 or newer). The algorithm to detect the percentage of positively stained pixels was tuned individually before analysis. Statistical analysis was performed using a Two-Way ANOVA, *<0.05, **<0.01, ***<0.001, ****<0.0001. Data were derived from two independent experiments (n = 8 in total; n = 4 per experiment). Black: uninfected; red: Delta-infected; blue: BA.1-infected (male animals, triangles; female animals, circles). Blue scale bar: 100 μm; green scale bar: 1 mm. Graphics made using biorender.com.

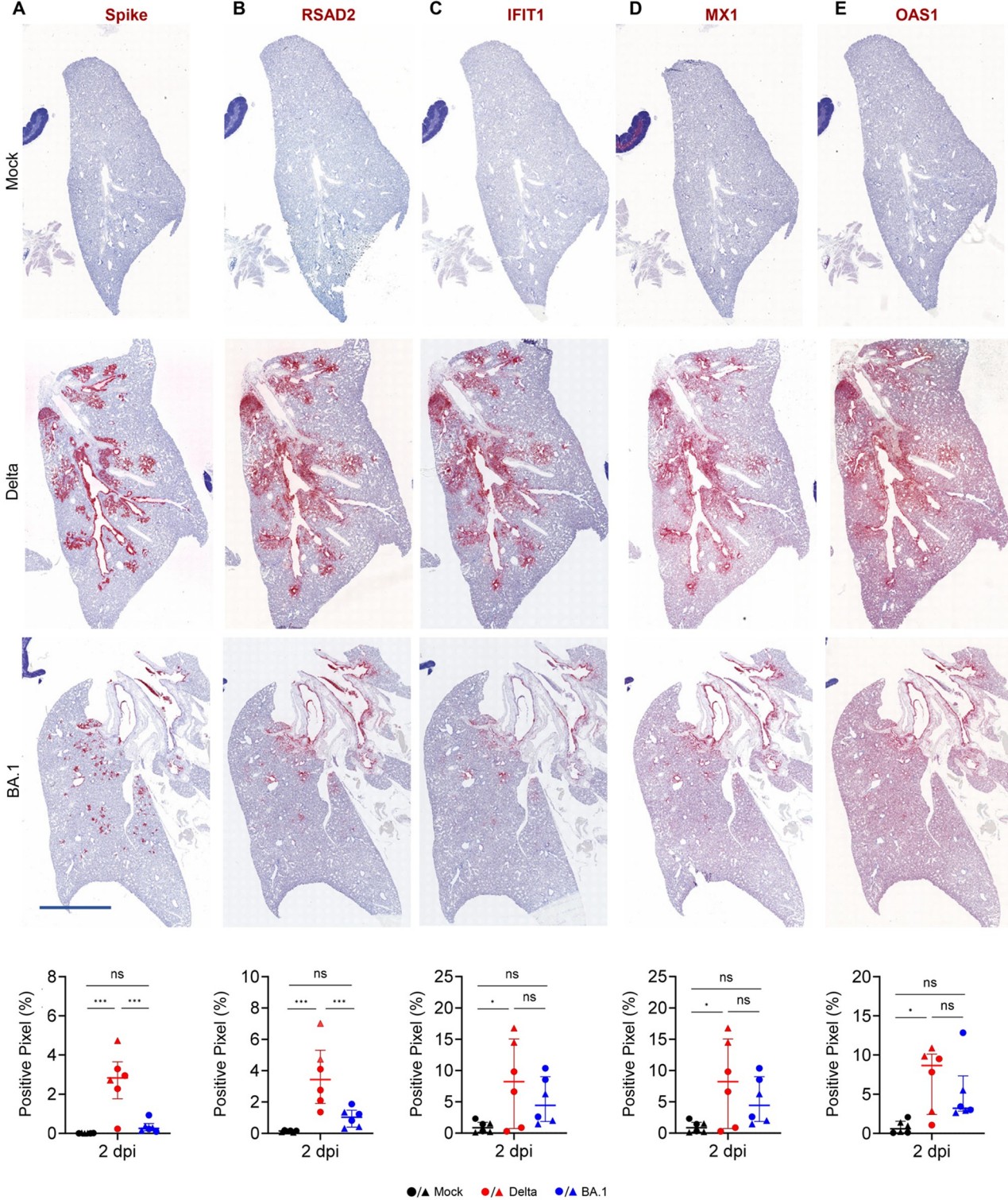

**Fig 3. Expression of interferon stimulated genes in lungs of experimentally infected hamsters.** (A) *In situ h*ybridisation of serial lung sections obtained from hamsters experimentally infected with either Delta or BA.1 and culled at 2 days post-infection (2 dpi). Probes used included those for SARS-CoV-2 spike, RSAD2, IFIT1, MX1 and OAS1. (B) Signal was quantified as in Fig 2 and statistical analysis was performed using a One-Way ANOVA, *<0.05, **<0.01, ***<0.001, ****<0.0001. Data represents two independent experiments using a total of n = 6 hamsters (3 females and 3 males per group). Black: uninfected; red: SARS-CoV-2 (Delta) infected; blue: SARS-CoV-2 (BA.1) infected (male animals, triangles; female animals, circles). Scale bar: 3 mm.

suggested that in addition to markers of the type-I IFN response, pathways leading to immune cell activation and tissue repair are also differentially upregulated between Delta- and BA.1-infected hamsters (Fig 1B and 1C). Histopathology lesions in the lungs of infected hamsters (especially those infected with Delta) were characterised by infiltration of macrophages in the alveoli and in the interstitium with a multifocal to coalescent distribution especially at 6 dpi (Figs 4, 5A and 5D). The immune cell infiltration also contained neutrophils/heterophils as well as lymphocytes and plasma cells (S4 Fig). At 2 dpi, Delta-infected hamsters showed vasculitis and a sloughing of bronchial epithelial cells. Vascular and bronchial lesions were instead minimal in BA.1-infected animals at 2 dpi. As shown in other studies [30, 42, 43], we found marked alveolar epithelial hyperplasia forming dense meander- and rosette-like structures replacing the alveolar spaces especially in the Delta-infected hamsters (Figs 4 and S4). The cellular infiltrates and the proliferation of the type 2 pneumocytes (or other lung progenitor cells), which starts around the bronchi involved a large area of the lung (especially in Delta-infected hamsters) (S4 Fig). Occasionally, multinucleated cells (interpreted as syncytia) (S5

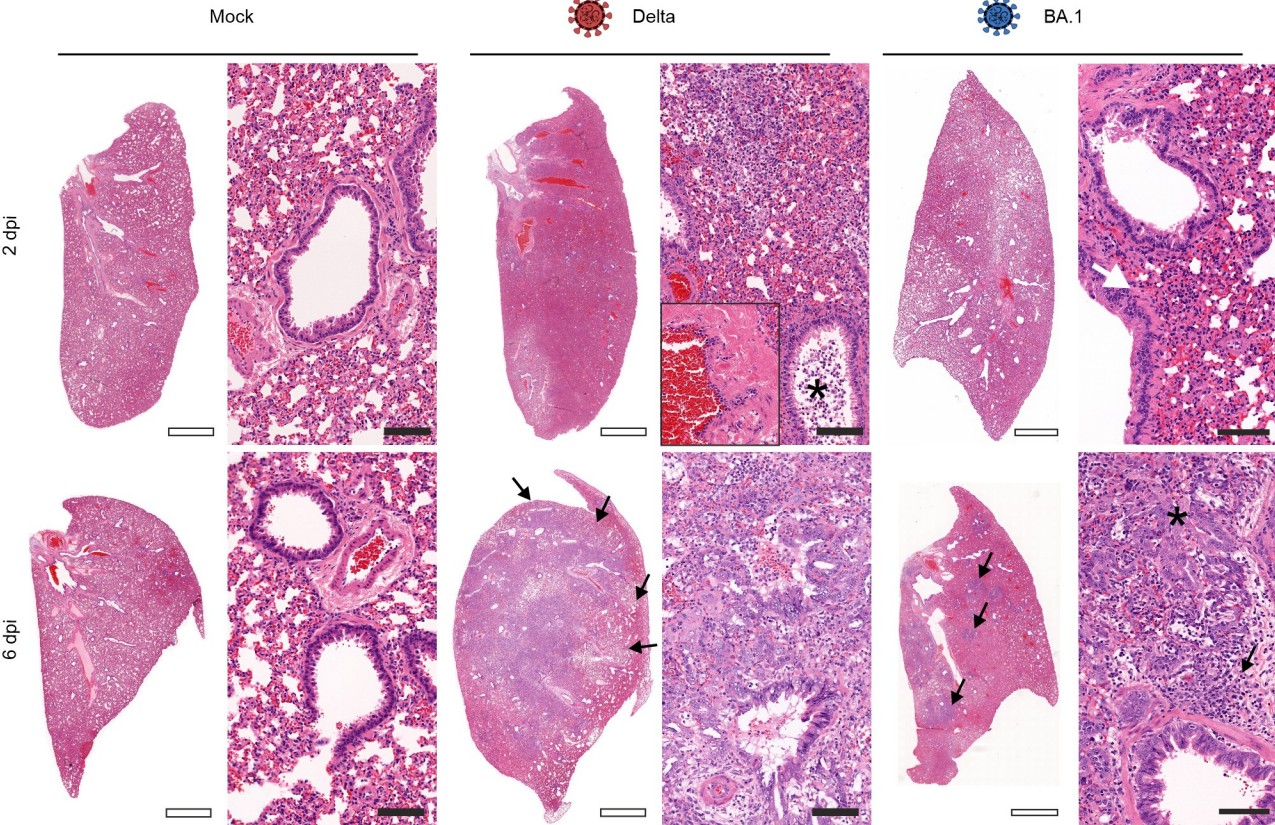

**Fig 4. Lung histopathology of SARS-CoV-2 experimentally infected hamsters.** Low (left) and high (right) magnification of lung sections stained with haematoxylin and eosin of hamsters experimentally infected with either the Delta or BA.1 (3.75x10$^6$ genome equivalent copies per animal), or mock-infected controls. At 2 days post-infection (dpi), Delta causes a higher level of infiltration of macrophages in and around bronchi, while BA.1 causes only mild infiltrations (white arrow) compared to mock-treated hamsters. On the same day, Delta-infected animals showed a moderate vasculitis (inset, right panel) and a moderate sloughing of bronchial epithelial cells (asterisk). Vascular and bronchi pathology was minimal in BA.1-infected animals at 2 dpi. At 6 dpi the lungs of Delta-infected animals show a severe infiltration of macrophages, lymphocytes, plasma cells and neutrophils/heterophils as well as a severe alveolar epithelial hyperplasia covering large areas of the lung lobe (black arrows; see also S4 Fig). BA.1-infected animals show in some cases the same lesions, including type 2 pneumocytes hyperplasia (asterisk) but covering only limited amounts of the lung lobes (black arrows). (Empty scale bars: 2 mm; filled scaled bars: 100 μm).

Fig) were associated with the often severely hyperplastic bronchial epithelium or in the lung parenchyma (Fig 4).

Hence, we next aimed to image and quantify the extent of virus-induced pulmonary pathology by first evaluating and comparing the immune cells infiltrate in the lungs of Delta- and BA.1-infected hamsters at 6 dpi. We specifically assessed T cells (CD3$^+$) and macrophages (IBA1$^+$) and found a significant increase in the number of these cells in Delta-infected hamsters compared to those infected with BA.1 (Fig 5A and 5B). By histopathology, both cell types represent most immune cell infiltrates in the lungs of Delta-infected hamsters (Fig 5A and 5B).

We next developed a method to quantify alveolar epithelial hyperplasia. We used the thyroid transcription factor (TTF1) [44, 45], a critical factor required for the expression of the surfactant protein in the respiratory epithelia. As expected, we found that TTF1-positive cells included the hyperplastic alveolar epithelial cells but also normal/isolated type 2 pneumocytes in the lungs and epithelial cells in the terminal bronchioles. To quantify only the hyperplastic areas in the lung parenchyma, we used software assisted imaging detection. Using supervised machine learning approaches, we "trained" the HALO software (Indica Labs) to detect clusters of TTF1-positive nuclear areas representing proliferating type-2-pneumocytes while ignoring isolated type 2 pneumocytes or TTF1$^+$ cells in the terminal bronchi. During the training, we made sure that the software excluded hyperplastic bronchial epithelium (S6 Fig). We found no hyperplastic lesions at 2 dpi in any of the hamster groups while at day 6 we found significantly more hyperplastic type 2 pneumocytes in Delta-infected hamsters compared to those infected with BA.1 (the latter had only values just above background in most animals; Fig 5C and 5D).

## Assessing the virulence of Omicron sub-lineages

The pipelines developed so far allow us to provide an automatic, unbiased, and quantitative method to assess the degree of virulence of SARS-CoV-2 in hamsters. Hence, we next used this method to assess virulence of recently emerged omicron sub-lineages.

We first investigated BA.5, as other studies, although carried out with chimeric BA.2/BA.5 viruses, suggested that this variant had an increased virulence compared to BA.1 [23]. Lungs collected from hamsters infected with BA.5 showed in comparison to those infected with BA.1, a consistent trend of increased values for (i) macrophage infiltrate (IBA-1$^+$ cells), (ii) cells expressing MX-1 and (iii) alveolar epithelial hyperplasia. Differences however lacked statistical significance (Fig 6A). BA.5-infected animals also showed a small decrease in body weight at 6 dpi, while animals infected with Delta or BA.1 showed the expected phenotype (slight increase in weight for BA.1-infected animals and weight loss for Delta from 2 dpi; Fig 6B).

Given that the digital pathology pipeline was able to show a possible intermediate virulence phenotype for BA.5, we proceeded to assess the virulence of other Omicron sub-lineages such as BQ.1.18, XBB and BA.2.75 (Fig 6C–6E). None of the variants induced major weight loss in infected hamsters (Fig 6C). Hamsters infected with BA.2.75 showed a modest weight loss between dpi 2 and 5 but neither hamsters infected with BA.2.75 nor BQ.1.18 show, unlike animals infected with BA.1, weight increase until 5 dpi (Fig 6C).

We found an increase in macrophages infiltrating the lungs in hamsters infected with the other variants compared to those infected with the original BA.1 (Fig 6D). BA.2.75 showed the highest levels of IBA1$^+$ cells in the lungs, although differences were statistically significant only with BA.1 and XBB, but not with BQ.1.18. Lungs of animals infected with BA.2.75 also displayed a significant increase of alveolar epithelial hyperplasia compared to lungs of BA.1-infected animals (Fig 6E). Overall, the data suggest that virulence of Omicron sub-lineages, and especially BA.5 and BA.2.75 has increased compared to BA.1. We also compared BA.2.75 to its

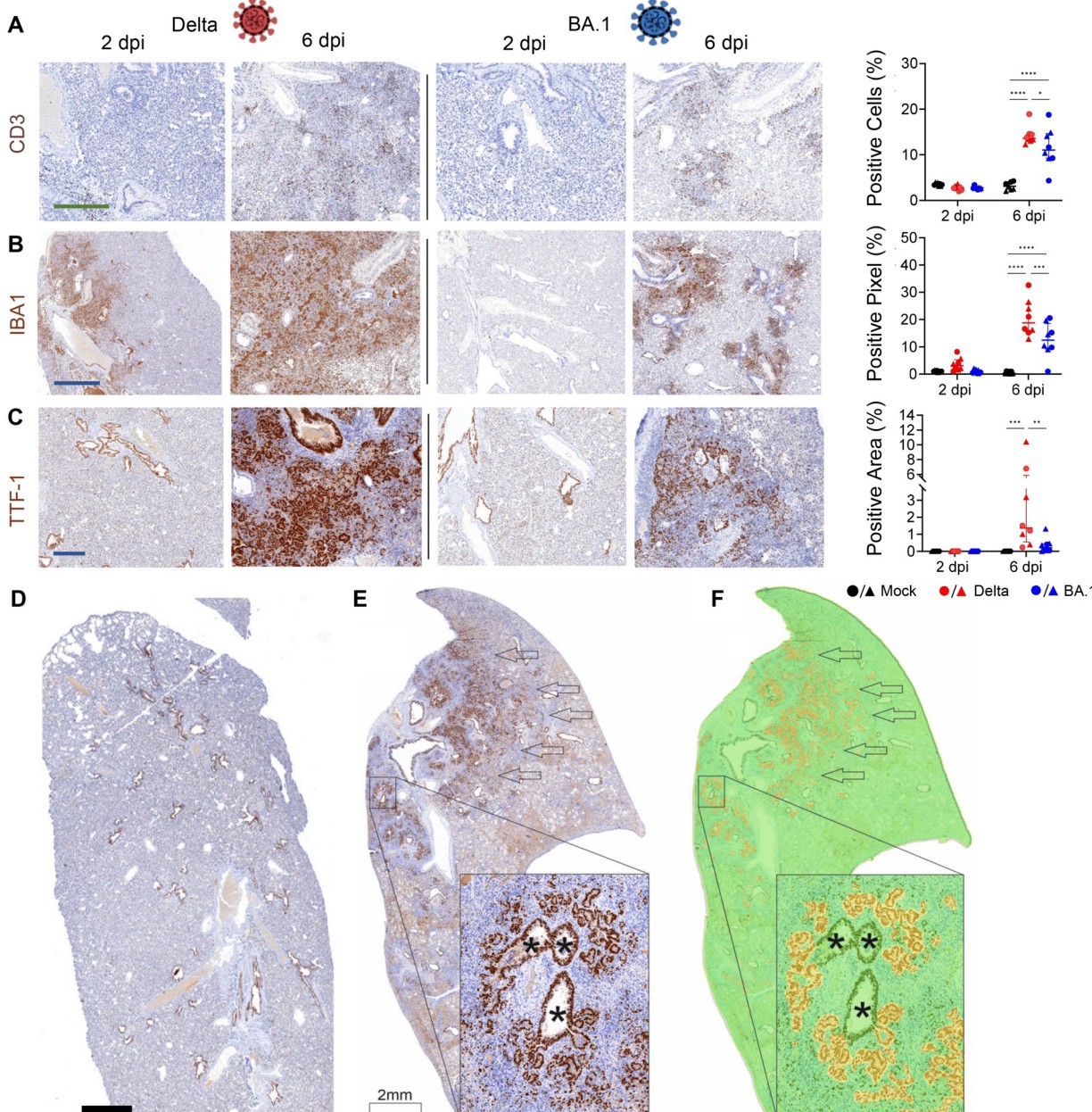

**Fig 5. Quantification of tissue histopathology in Delta or BA.1-experimentally infected hamsters.** Images of whole lung sections of hamsters experimentally infected with either Delta or BA.1 ($3.75 \times 10^6$ genome equivalent copies per animal) or mock-infected controls, and culled at either 2- or 6-days post-infection (dpi). Expression of CD3 (A), IBA1 (B) and TTF1 (C-D) was assessed by immunohistochemistry. Animals were culled 2- or 6-days post infection (dpi) and the lungs were collected for histological analysis. (A-C) Representative photomicrographs for each marker and associated quantification are shown. (D) Mock-infected animals show a positive signal for TTF1 only in bronchial epithelium and in single cells scattered throughout the lung parenchyma. (E) Lungs of Delta-infected hamsters shows severe proliferation of TTF1+ positive cells (arrows). The inset shows a higher magnification of type 2 pneumocytes hyperplasia surrounding centrally located bronchi (asterisk). (F) Same image as in E after analysis with the HALO software algorithm trained to detect alveolar epithelial hyperplasia. The software shows negative areas in green and positive ones in yellow; only proliferating alveolar epithelial cells are identified as positive areas, while TTF1+ type-2 pneumocytes and bronchial epithelial cells are excluded by the software. CD3- and IBA1-positive cells in experimentally infected animals were quantified using QuPath (Version 0.3.2 or newer), while HALO was used to quantify alveolar epithelial hyperplasia. Statistical analysis was performed using a Two-Way ANOVA, *<0.05, **<0.01, ***<0.001, ****<0.0001. Data were obtained from two independent experiments using 8 hamsters (4 females and 4 males infected in two independent experiments). Black: uninfected; red: Delta-infected; blue: BA.1-infected (males, triangles; females: circles). Blue scale bar: 1 mm; green scale bar: 400 μm.

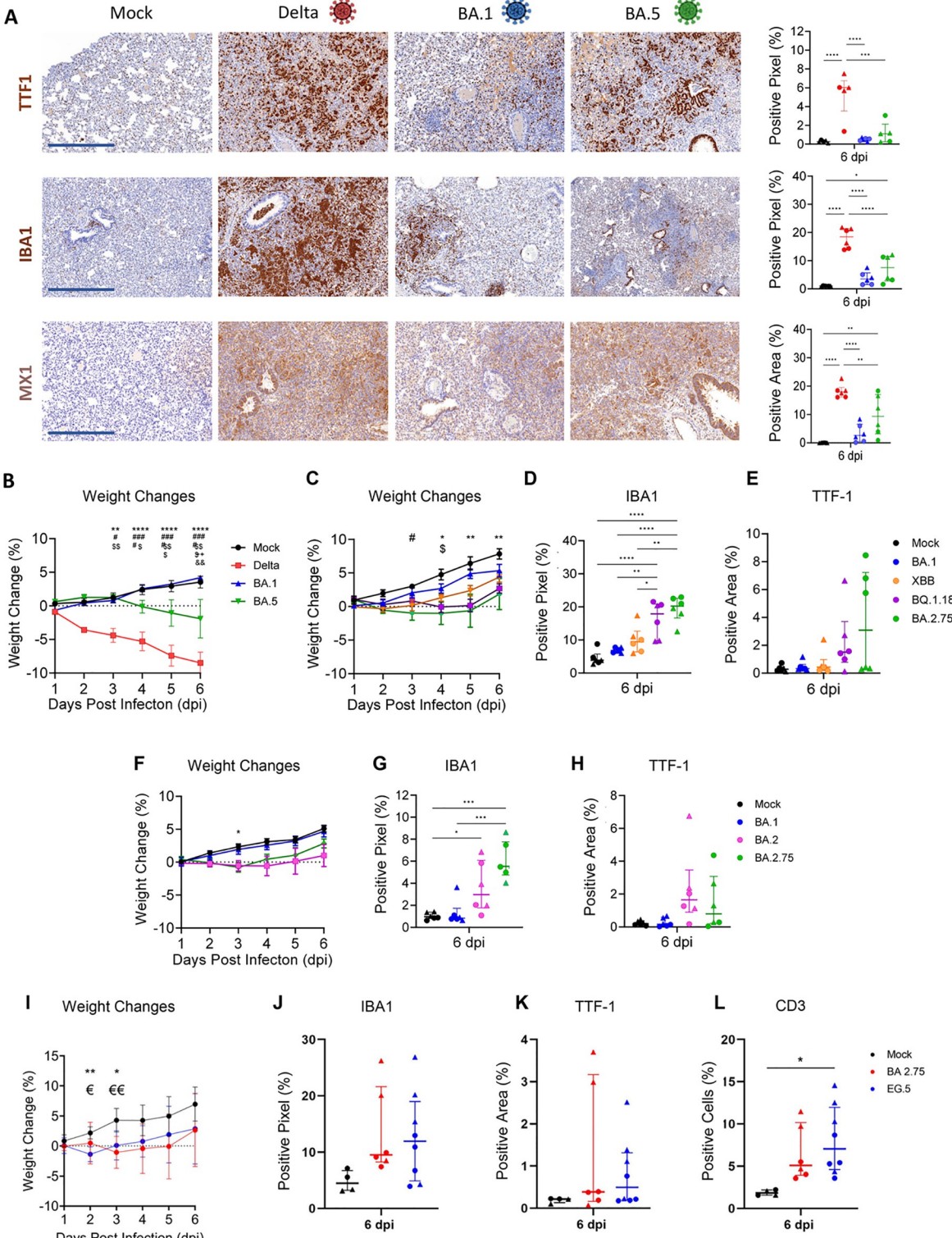

**Fig 6. Quantification of Omicron sub-lineages virulence in experimentally infected hamsters.** (A) Photomicrographs of lung sections collected at 6 days post-infection (dpi) from hamsters experimentally infected with either Delta, BA.1, BA.5 (virus load equivalent to 3.75x10$^6$ genome copies per hamster) or mock-infected. Expression of TTF1$^+$ hyperplastic epithelial cells and infiltrating macrophages (IBA1$^+$) was assessed by *in situ*-hybridisation. Expression of MX1 was instead assessed by immunohistochemistry. Data were quantified as in Fig 5. (B) Differences in the weights between these animals were recorded daily. (C) Daily recorded weight changes in animals mock-

infected or infected with either BA.1, XBB, BQ.1.18 or BA.2.75. (D) Lung whole scanned sections were analysed for the presence of IBA1+ cells or (E) hyperplastic alveolar epithelial cells as in A. (F) Daily weight changes of mock-infected hamsters or animals infected with either BA.1, BA.2 or BA.2.75. (G) Quantification of cells expressing IBA1 or (H) hyperplastic TTF1+ cells. Comparison of weight loss (I), macrophage infiltration (J), alveolar epithelial hyperplasia (K) and T cells infiltration (L) in the lungs of BA2.75 or EG.5.1-infected hamsters. Statistical analysis was performed using a One-Way ANOVA with Tukey's multiple comparisons test. Weight comparisons were performed using a Two-Way ANOVA. Data are shown as mean +/- standard deviation. Significance is indicated with *<0.05, **<0.01, ***<0.001, ****<0.0001. * Denote comparisons between Delta and mock, or BQ.1.18 and mock, or BA.2.75 and mock; # denote comparisons between BA.1 and delta, or mock and BA.2.75; $ denotes comparisons between Delta and BA.5, or mock and XBB; + denotes comparisons between uninfected and BA.5; & denotes comparisons between BA.5 and BA.1. n = 6 (3 females and 3 males per group). **€ denotes the differences between mock and EG.5.1, and *€€ denotes differences between mock and BA2.75, as well as mock and EG.5.1. Black: uninfected; red: Delta-infected; blue: BA.1-infected; green inverted triangles: SARS-CoV-2 (BA.5) infected; orange inverted triangles: SARS-CoV-2 (XBB) infected; purple squares: SARS-CoV-2 (BQ.1.18) infected; pink squares: SARS-CoV-2 (BA.2) infected; green diamonds: SARS-CoV-2 (BA.2.75) infected. Males: triangles, females: circles. Scale bar: 500 μm. Graphics made using biorender. com.

predecessor BA.2 to determine whether there had been further evolutionary adaptations in BA.2.75 that favoured a more virulent phenotype. No significant differences were observed for weight losses, amount of macrophages and alveolar epithelial proliferation between BA.2.75 and BA.2 (Fig 6F–6H). However, animals infected with BA.2.75 or those infected with BA.2 did not gain weight as steadily as mock- or BA.1-infected animals. We saw however a trend for BA.2.75-infected hamsters to show an increase in the levels of infiltrating macrophages in the lungs compared to BA.2, although differences were not statistically significant (Fig 6G). All variants induced limited levels of alveolar epithelial hyperplasia (Fig 6H), in line with the limited ability of these viruses to invade the lung parenchyma. Finally, to visualise and compare all the different variants used in this study, we normalised the data using values obtained in BA.1-infected hamsters as unit of reference between different experiments (S7 Fig). Graphs displayed in S7 Fig show, as expected, a gradient of virulence between variants. XBB induced little or no lung pathology similarly to BA.1. BA.5, BA.2 and BQ.1.18 were more virulent than BA.1, while BA.2.75 was clearly more virulent than BA.1 but not as virulent as Delta. The use of a standard virus of reference could therefore enable comparisons between variants used in different experiments (and different laboratories).

During the revision of this manuscript, the spread of EG.5.1 around the world substantially increased [9]. We compared the virulence of EG.5.1 to BA.2.75, as the latter showed relatively higher virulence in hamsters compared to other Omicron-like viruses. We found the virulence of EG.5.1 to be significantly higher than BA.1 but comparable to BA.2.75 (Fig 6I–6L).

For comparative purposes, semiquantitative scoring [39] of the lung pathology was also performed simultaneously by three board certified pathologists on the lungs of hamsters infected with BA.1, XBB, BQ.1.18 and BA.2.75. Overall data showed a similar trend compared to our digital pathology pipelines (S8 Fig).

## Discussion

The COVID-19 pandemic has entered a phase characterised by the periodic emergence of immune escape variants, which may differ in their virulence from their predecessors. Assessing the virulence of any newly emerged variant will be therefore one of the key features requiring near real-time monitoring. Animal models have been used throughout the pandemic to unveil many aspects of SARS-CoV-2 pathogenesis [43]. So far, there has been an excellent correlation between the virulence of SARS-CoV-2 variants such as Delta and BA.1 in humans and in experimentally infected hamsters [7, 18–21, 26, 27, 42, 46, 47].

In many studies, virulence of SARS-CoV-2 variants in experimentally infected hamsters has been determined by assessing body weight loss, and various features of lung function and

histopathological lesions [8, 15, 17, 18, 23–27, 48]. In general, these parameters have proven to be good proxies of the virulence of variants. This is particularly so for variants such as Delta and Omicron, which exhibit clearly distinct phenotypes with respect to virulence. Some relative discrepancies between studies can, however, arise when the intrinsic differences in virulence between variants are modest, as for example with BA.2 and BA.5 compared to BA.1 [23, 26].

Histopathological features in infected animals such as alveolar damage and inflammatory lesions in the respiratory tract are a consequence of virus replication (and host immune responses) and therefore directly reflect virus virulence. In various studies, lung pathology is often characterised by qualitative scores determined by trained pathologists on lesions such as bronchiolitis, haemorrhages, alveolar damage and others (S9 Fig). Qualitative histopathology scores can vary between individuals, and therefore these types of data are difficult to compare unequivocally between different laboratories. Implementation of digital-pathology pipelines assisted by machine learning not only increase objectivity by eliminating inter- and intra-observer variability, but it also increases the speed of data analysis [49]. In addition, assessing lesions in a semi-quantitative fashion with a limited number of options (i.e. scoring with 0, 1, 2, 3) may mask the variability between individuals that can be better appreciated with a more quantitative method used in this study. In this study, we aimed to develop a framework for quantitative unbiased methods to phenotype the relative virulence of SARS-CoV-2 variants by quantifying the extent of pulmonary pathology in experimentally infected hamsters. In line with previous studies [39, 50–52], our transcriptomic analysis showed that SARS-CoV-2 infection induces an infiltrate of immune cells in the lungs and immune activation in general. Indeed, here we found that expression of ISGs, and infiltrate of T cells and macrophages are common in lesions induced by the hypervirulent variant Delta but are barely above background in BA.1-infected hamsters.

In addition, our RNAseq data suggest that tissue remodelling was another key parameter distinguishing the transcriptional profile of Delta- and BA.1-infected hamsters. Virulent SARS-CoV-2 variants induce alveolar damage, with necrosis of type 1 and type 2 pneumocytes. Lung damage is subsequently repaired by alveolar epithelial hyperplasia, which is a key feature of lesions induced by SARS-CoV-2 in experimentally infected hamsters [30, 42, 43], and it is also found in some cases in post-mortem samples of human patients dying as result of COVID-19 [53–55]. Lung respiratory epithelium repair is due to proliferation of either type 2 pneumocytes following mild injuries [56], or other lung progenitor cells in response to severe injury with abundant loss of type 1 pneumocytes [57–59].

Overall, we found that infiltrating macrophages and proliferating epithelial cells constitute most of the cells in the affected areas of the lung parenchyma of hamsters infected with virulent SARS-CoV-2 variants. We consistently observed epithelial hyperplasia and macrophage infiltrates also in a separate study based on histopathology of the lungs of patients who died as result of the first wave of COVID-19 [60]. These two features represent therefore exceedingly good markers that by themselves provide an unbiased quantification of the virus-induced pulmonary lesions, and by extension virus virulence. In addition, the use of whole lung sections can also reduce bias by providing a spatial overview of the inflammatory response. For example, our investigation of the spatial distribution of the ISG response and how it directly correlated to the presence of virus in an inflammatory lesion proved to be particularly insightful (Fig 3). Whole scanned imaging of tissue sections and downstream analyses including those based on artificial intelligence approaches ("digital pathology") have been used extensively in diagnostic pathology and other fields including cancer research and infectious diseases [49, 61, 62]. Artificial intelligence-based pipelines to diagnose lung pathology in humans have emerged recently [63]. In these approaches, the software is trained

to distinguish between different diseases such as tuberculosis and lung cancer as well as fibrotic conditions.

In our study, by performing immunohistochemistry of whole scanned lungs sections to identify IBA1$^+$ cell (as convenient marker for macrophages), we were able to quantify infiltrating macrophages in the lung parenchyma of experimentally infected hamsters. In this manner, the relative number of infiltrating macrophages in the lung parenchyma can be acquired in an unbiased fashion. The same approach to measure alveolar epithelial hyperplasia by simply quantifying TTF1$^+$ cells did not initially provide satisfactory results, due to the expression of this marker in both proliferating and non-proliferating type 2 pneumocytes, as well as bronchial epithelial cells. However, supervised machine learning approaches allowed us to train the software used to detect clusters of TTF1$^+$ cells (hyperplastic type-2 pneumocytes) and exclude isolated type 2 pneumocytes (representing the normal type 2 cells in the lungs) or TTF1$^+$ cells in the bronchiolar epithelium. Our approach allowed us not only to clearly distinguish the pulmonary lesions between those induced by the virulent Delta and the attenuated BA.1, but also to show an increased virulence of other Omicron sub-lineages.

The pipeline used in this study, enabled us to show that BA.5, BA.2.75 and EG.5.1 have acquired an increased virulence phenotype compared to BA.1, as also suggested in other recent studies [6, 23, 27]. We saw no differences instead in virulence between XBB and BA.1, also in keeping with another recent study [8]. Interestingly we found a tendency for BA.2 to induce a higher number of macrophage infiltrates and type 2 pneumocyte hyperplasia compared to BA.1, although differences did not reach statistical significance. Other published studies found no major differences in virulence between BA.1 and BA.2 [8, 48], in contrast to a previous study which used recombinant viruses with either BA.1 or BA.2 spike [64].

We propose that the approach described in this study can be used to quantify moderate differences in virulence between variants, although animal group sizes may need to be adjusted to address specific experimental questions. Indeed, most studies focusing on SARS-CoV-2 virulence in hamsters use experimental groups between 4 and 6 animals (and often males only), but these may not be sufficient to reveal relatively modest differences in virulence between variants and the inherent individual variability (considering also that scores are set with values between 0 and 3 or 4). It may be argued that modest differences in variant virulence observed in hamsters could have limited biological significance in human patients. Indeed, a limitation of these type of studies is that the intrinsic virulence of variants in naïve hamsters is difficult to compare to their virulence in the "real world" in the human population (with pre-existing immunity derived from vaccination or previous infections). However, the intrinsic virulence of newly emerging variants remains a key phenotype to monitor to determine whether adjustments to public health measures are needed. For example, the emergence of a variant with increased virulence may require different vaccination policies from the existing ones.

We use equal number of both male and female hamsters in all our experiments. As expected, we observed some gender-independent variability in the extent of the lesions caused by the same variant between individual hamsters both within and between different experiments. A standard reference virus could be used in multiple experiments assessing different variants to normalise the data between experiments (S7 Fig). This approach would also enable meaningful comparison of data between different laboratories.

In conclusion, the digital pathology pipelines developed in this study provide a framework for the quantitative assessment of virulence of SARS-CoV-2 variants in experimentally infected hamsters. The identification of virus standards with defined pathogenicity and sharing of protocols for the quantification of pulmonary lesions, can allow comprehensive comparison of *in vivo* data between different laboratories. We also developed an online repository to share more effectively with the research community histopathological images derived from scanned whole

organ tissue sections. We argue that the reader will be able to appreciate better histopathological micrographs contained in the main body of manuscripts if these were supplemented by images of whole scanned slides similarly to those contained in our virtual microscope. Due to space constraints, micrographs contained in figures of standard manuscripts show only individual areas of interest at single or two magnifications. Virtual microscopes allow the reader to view all areas of a given section and at multiple magnifications, providing therefore a comprehensive spatial context of the data.

## Materials and methods

### Ethics statement

Animals were maintained at the University of Glasgow under specific pathogen free conditions in accordance with UK Home Office regulations (Project License PP0271643), in accordance with the Animals (Scientific Procedures) Act 1986, and approved by the University of Glasgow ethics committee. All animal research adhered to ARRIVE guidelines.

### Cells and viruses

Calu-3 cells (ATCC HTB-55) are human lung adenocarcinoma epithelial cells. African green monkey kidney cells (Vero E6) expressing the human Ace2 receptor [65] were used to propagate the viruses. All cell lines were maintained at 37˚C and 5% $CO_2$ in DMEM (ThermoFisher) supplemented with 10% foetal bovine serum (FBS) (ThermoFisher), except for Calu-3 cells where RPMI-1640 medium (ThermoFisher) was supplemented with 20% FBS. BA.5 virus isolate was obtained as passage P2 from Greg Towers (University College London, UK) after initial access to passage P1 obtained from Alex Sigal (AHRI, South Africa). Other variants used in this study included the Delta variant (B.1.617.2, GISAID accession number EPI_ISL_1731019), BA.1, BA.2, BA.2.75, BQ.1.18, and XBB all isolated from clinical samples obtained either at the CVR or at the Imperial College London. Viruses were isolated from a clinical nasopharyngeal swab sample collected in virus transport medium. Samples were then resuspended in serum-free DMEM supplemented with 100 units mL-1 penicillin-streptomycin, 10 ug mL-1 gentamicin and 2.5 ug mL-1 amphotericin B to a final volume of 1.5 mL. Calu-3 cells were then inoculated and incubated overnight at 37C, 5% CO2. The next day, cell culture medium was replaced with fresh complete medium before further incubation. Infected cells were monitored for signs of CPE and the presence of viral progeny in supernatant by RT-qPCR. All working stocks were propagated in VeroE6 cells.

### Animals

Male and female golden Syrian hamsters (HsdHan:AURA) were bred and maintained by Envigo; Wyton, United Kingdom). Animals were shipped as required and acclimatised at the Veterinary Research Facility (VRF) at the University of Glasgow prior to transfer to Containment Level 3 (CL3) suite. Animals were maintained in individually ventilated cages (IVCs) on a 12-hour light/dark cycle and provided with food and water *ad libitum*. Both male and female hamsters between 8–12 weeks old were used for experiments.

### Experimental infections

Animals were randomised to treatment groups, however, due to the nature of this work blinding was not possible. Animals were handled within a Class I microbiological safety cabinet (MSC) in a CL3 suite throughout the experiment. Hamsters were anaesthetised with oxygen (1.5 L/minute) containing 5% isoflurane and intranasally dosed with 50 μl of Dulbecco's

Modified Eagle Medium (DMEM) containing $3.75 \times 10^6$ genome copies (the equivalent of approximately $1 \times 10^4$ PFU) of SARS-CoV-2. We chose to use genome copies for our infection studies to control for the differences in infectivity and plaque formation that have been observed with the BA.1 variant [16]. Control animals received DMEM only. Animal weights and temperatures were recorded daily, and animals were monitored twice daily for signs of clinical disease including piloerection, hunching, abnormal breathing and reduced peer interactions. Animals received a disease score based upon weight loss and the presence and severity of these signs. Animals were culled at the end of the experiment via a rising concentration of $CO_2$.

## Throat swabs

Animals were restrained and swabs (MWE) were inserted into the mouth and rotated five times on each tonsil. The swabs were then placed in 2 ml DMEM (ThermoFisher) containing 2% FBS (ThermoFisher) for 1 minute. For RNA RT-qPCR, 250 μl of media was added to 750 μl TRIzol LS (ThermoFisher). For live virus assays, the media was frozen at -80˚C.

## RNA extractions

Viral RNA was extracted from culture supernatants using the RNAdvance blood kit (Beckman Coulter Life Sciences) following the manufacturer's instructions. Tissue samples, approximately 20 mg in size, were collected in homogenisation tubes containing 2.8 mm metal beads (Stretton Scientific) and 1 ml TRIzol reagent (ThermoFisher Scientific). Blood was collected in TRIzol LS reagent (ThermoFisher Scientific) at a 1:4 ratio. Tissue samples were homogenised (6500 rpm, 4x 30s cycle, 30s break, room temperature) using a Precellys Evolution Homogeniser (Bertin Instruments) and the homogenate was mixed with 200 μl chloroform (Merck—1L). Each sample was mixed thoroughly and centrifuged at 12,000 x g for 15 minutes at 4˚C. The aqueous phase was removed and mixed with 230 μl 100% ethanol (Merck). The samples were then added to columns from RNeasy Mini Kits (Qiagen) and processed as per manufacturer's instructions. On column DNase digestion (Qiagen) was performed on all samples for 15 minutes as per manufacturer's instructions.

## RT-qPCR

RNA was used as template to detect and quantify viral genomes by duplex reverse transcriptase (RT) quantitative polymerase chain reaction (RT-qPCR) using a Luna Universal Probe one-step RT-qPCR kit (New England Biolabs, E3006E). SARS-CoV-2-specific RNAs were detected by targeting the ORF1ab gene using the following set of primers and probes: SARS-CoV-2_Orf1ab_Forward 5'GACATAGAAGTTACTGGCGATAG3', SARS-CoV-2_Orf1ab_Reverse 5'TTAATATGACGCGCACTACAG3', and SARS-CoV-2_Orf1ab_Probe 5' HEX-ACCCCGT-GACCTTGGTGCTTGT-BHQ-1 3'. Hamster β-actin was used as a reference gene using the following primers hACTB-F 5′CTCCCAGCACCATGAAGATC3′, hACTB-R 5′GCTGGAAGGTGGACAGTG3′, and hACTB-Probe 5′ Cy5-TGTGGATCGGTGGCTC-CATCCTG-BHQ-3 3′. Normalisation was performed using the ΔΔCt method and SARS-CoV-2 genomic copies were calculated by interpolating adjusted ORF1ab Ct values from a standard curve. All runs were performed on the ABI7500 Fast instrument and results analysed with the 7500 Software v2.3 (Applied Biosystems, Life Technologies).

## Histology, immunohistochemistry and *in situ* hybridization

3 μm thick sections of formalin-fixed (8%) and paraffin-wax embedded (FFPE) tissues (lung, trachea, larynx, and EDTA-decalcified turbinates) were cut and mounted on glass slides. We ensured that lungs were always placed in the cassette in the same orientation before the embedding process. Whole organ horizontal sections were also cut at the same level for each animal. Slides were stained with haematoxylin and eosin (HE). The following antibodies were used: anti-SARS CoV-2 nucleocapsid (Novus Biologicals), anti-CD3 (Agilent DAKO) anti-Pax5 (Agilent DAKO), anti-TTF-1 (Leica), anti-E-cadherin (Agilent DAKO), anti-IBA1 (Fujifilm), and anti-MX1 (Cell Signaling). As negative control, the primary antibody was replaced by isotype serum. For visualization, the EnVision+/HRP, Mouse, HRP kit (Agilent DAKO) or EnVision+/HRP, Rabbit, HRP kit (Agilent DAKO), respectively, was used in an automated stainer (Autostainer Link 48, Agilent Technologies). For all immunohistochemistry experiments, 3,3'-Diaminobenzidine (DAB) was used as a chromogen. RNA was detected using RNAScope according to manufacturer's instructions (Advanced Cell Diagnostics, RNAScope) with simmering in target solution and proteinase K treatment. The following probes were used: SARS-CoV-2 spike (product code: 848561), DapB (product code: 310043), Ubiquitin (product code: 310041) as well as cricetine MX1 (product code: 1153131), IFIT1 (product code: 1153111), OAS1 (product code: 1153101), RSAD2 (product code: 1153121).

## Digital pathology: Standard signal quantification

For image analysis, slides were scanned with an Aperio VERSA 8 Brightfield, Fluorescence & FISH Digital Pathology Scanner (Leica Biosystems) at 200 x brightfield magnification [66].

To quantify the positive signal of Fast-red (*in situ*-hybridisation, ISH) and diamino-benzidine (DAB)-brown (immunohistochemistry, IHC) in the tissues (nose, turbinates, larynx, trachea, and lung), we optimised image analysis pipelines based on the signal detected using the QuPath software (version 0.3.2 or later). The initial step was to upload and set the appropriate image type [Brightfield (other) for ISH; Brightfield (H-DAB) for IHC] and manually outline the area of interest using the 'wand and 'brush' tools. Staining obtained for SARS CoV-2 spike RNA (ISH), IBA1 (IHC), MX1 (IHC) as well as various ISGs including IFIT1, MX1, RSAD2 and OAS1 (ISH), we utilised the pixel classifier feature within QuPAth. This specific (pixel classifier) was selected for analysis as, in these stainings, the virus (spike RNA) signal was very abundant, with multiple positive cells being clustered together or in patches. Conversely, for the ISGs, the positive signal was dispersedly distributed with a punctuate pattern. In detail, for each Pixel detection (%), the image type was set as described above. To create the thresholder, the 'Resolution' was set to 1.09 μm/pixel or higher. For the 'Channel', we used 'Red' (ISH) or 'DAB' (IHC); the 'Prefilter' was always 'Gaussian' while 'Smoothing sigma' as well as the 'Threshold' have been tuned for each set of experiments to optimise the correct detection (based on each individual batch of slides with their individual staining intensities and background). The 'Above threshold' option was always set to 'Positive' whereas the 'Below threshold' was always set to 'Negative'. The 'Region' to be analysed was set to 'Any annotation ROI'. The readout is a percentage of positive pixels per total pixels in the annotated area.

The 'Positive cell' detection feature has been used instead for the detection of cell membrane-associated CD3-positive cells. The settings for the 'Requested pixel size' were adjusted to 0.5 μm or smaller and the 'Nucleus parameters' included a 'Background radius' of 8 μm. The 'Median filter radius' was set to 0.8 μm or smaller, 'Sigma' to 1.5 μm or smaller while the 'Minimum' and 'Maximum' areas were adjusted to 10–30 μm$^2$ and 400 μm$^2$ respectively. The 'Intensity parameters' included a 'Threshold' of 0.1 or 0.2 and a 'Max. background intensity' of 2.

The 'Cell parameters' included a 'Cell expansion' of 5 μm. The readout is a percentage of positive cells detected per all cells in the annotated area.

## Digital pathology: Quantification of alveolar epithelial hyperplasia

We used the algorithm 'AI' within the software HALO (Indica Labs) to detect clusters of TTF1$^+$ nuclei within the lung corresponding to alveolar epithelial hyperplasia, while ignoring individual TTF1$^+$ normal type 2 pneumocytes, and the positively stained normal or hyperplastic bronchial epithelium. The readout is therefore the percentage of positive TTF1-stained area in the lungs with rosette-like shapes. First, the software was trained to detect positive areas of interest. The files with the trained algorithm to detect alveolar epithelial hyperplasia are shared in the Enlighten repository (https://doi.org/10.5525/gla.researchdata.1513) and can be imported to any HALO software (HALO version 3.4.2986.230) by the users. The file ".classifier" can be read by a text editor and includes metadata (resolution, class names, etc.). The ".params" and ".trainer" files are specific to the deep learning framework mxnet. The.params files MXNet's specific file format for storing model weights. The.trainer files are MXNet's specific file format for the optimizer. This keeps track of training information such as iteration, learning rate and optimizer states. A zip file with these three files can be loaded directly into HALO.

The HALO algorithm (Densenet AI V2 plugin) was trained on 3 sets of independent experiments. Each group of animals were handled as separate batches and infected, processed, stained and scanned in 3 different times as independent experiments. The samples were processed in the same way across multiple experiments using the same machine learning algorithm. We selected at least 30–40 slides across 3 different experiments containing delta and omicron-infected lung samples to train the software and optimize its performance. First, regions of interest (ROIs) were selected on each slide. The DenseNet AI plugin classifier was used, and classes were defined for positive (TTF-1 positive and proliferating type 2 pneumocytes) and negative tissue (isolated type 2 pneumocytes, in addition to normal or hyperplastic bronchial epithelium). Example regions were drawn within each class at high magnifications, and any significant TTF1-negative cells within the rosette structures were excluded. After example regions of epithelial hyperplasia were drawn, the classifier was trained over multiple days. Raw data is expressed as the total area covered by the rosette structures (TTF-1 positive nuclei, alveolar epithelial hyperplasia), in comparison to the overall tissue area. Slides with artefacts were excluded from the analysis as well as artefacts on the slides, which have been manually excluded. Additional examples of the performance of the software algorithm developed in this study to detect alveolar epithelial hyperplasia are shown in S6 Fig. Additionally, we also compared data related to quantification of alveolar epithelial hyperplasia from two different lung sections (approximately 100μm apart) from each animal experimentally infected with either BA.1, BA.2 or BA.2.75 (S9 Fig) The data obtained with either section (termed for simplicity "section 1" and "section 2") were very similar. Therefore, we used a single section per animal in subsequent experiments.

All photomicrographs have been captured with Aperio ImageScope (Leica Biosystems).

Accuracy of the training outputs was confirmed by a Board certified veterinary pathologist. For some experiments, the data obtained with the digital pathology pipeline was further compared to semiquantitative scoring of HE-stained lung sections by three board-certified pathologists in a blinded fashion according to previously published studies [39]. Briefly, lesions (amount of lung affected, respiratory epithelial cell or type II pneumocyte hyperplasia, alveolar damage, perivascular inflammation, damage in bronchi/bronchioles) were scored from 0 to 4 for each lung (n = 6 hamsters) by each pathologist (n = 3). Data are also shown as a cumulative

score containing all 5 parameters from all pathologists. In this case, results are shown as the sum of the average value scored by each pathologist for each animal.

## Online digital pathology tool

Representative whole scanned images are available online at https://covid-atlas.cvr.gla.ac.uk. The "CVR Virtual Microscope," is an online tool where users can zoom in and out of digital images of scanned tissues, accessing therefore the same context and information experienced on a microscope.

## RNA sequencing (Bulk RNAseq)

Total RNA extracted from lung and blood samples from uninfected and infected animals was quantified using Qubit Fluorometer 4 (Life Technologies; Q33238), Qubit RNA HS Assay (Life Technologies; Q32855) and dsDNA HS Assay Kits (Life Technologies; Q32854). RNA integrity number was determined on a 4200 TapeStation System (Agilent Technologies; G2991A) using a High Sensitivity RNA Screen Tape assay (Agilent Technologies; 5067–5579). Before library preparation, haemoglobin RNA was removed from 1 μg of RNA extracted from blood samples using the GLOBINclear-Mouse/Rat Globin mRNA Removal Kit (Thermo Fisher Scientific; AM1981) following the manufacturer's instructions. Total RNA (500 ng) was used to prepare libraries for sequencing using the Illumina TruSeq Stranded mRNA Library Prep kit (Illumina; 20020594) and SuperScript II Reverse Transcriptase (Invitrogen; 18064014) according to the manufacturer's instructions. The PCR amplified dual indexed libraries were cleaned up with Agencourt AMPure XP magnetic beads (Beckman Coulter; A63881), quantified using Qubit Fluorometer 4 (Life Technologies; Q33238) and Qubit dsDNA HS Assay Kit (Life Technologies; Q32854). Their size distribution was assessed using a 4200 TapeStation System (Agilent Technologies; G2991A) with a High Sensitivity D1000 Screen Tape assay (Agilent Technologies; 5067–5584). Libraries were pooled in equimolar concentrations and sequenced using high output cartridges with 75 cycles (Illumina; 20024911) on an Illumina NextSeq 550 sequencer (Illumina; SY-415-1002). A Q score of ≥30 was presented in at least 95% of the sequencing reads generated.

## Sequence quality and assembly

Prior to performing bioinformatics analysis, RNA-Seq reads quality was assessed using FastQC software (http://www.bioinformatics.babraham.ac.uk/projects/fastqc). Sequence adaptors were removed using TrimGalore (https://www.bioinformatics.babraham.ac.uk/projects/trim_galore/). Subsequently, the RNA-Seq reads were analysed. Sequence reads were aligned to the *Mesocricetus auratus* genome (MesAur1.0), downloaded via Ensembl using HISAT2. HISAT2 is a fast and sensitive splice-aware mapper, which aligns RNA sequencing reads to mammalian-sized genomes using the FM index strategy [67].

## Differential expression genes analysis

After the alignment to the *Mesocricetus auratus* genome, FeatureCount [68] was used to calculate the mapped reads counts. In this paper, we observed the differential expression genes (DEGs) of mock vs Delta and mock vs BA.1 (2/6 days) on both lung and blood cells. The DESeq2 [69] in Generalized linear models (GLMs) with multi-factor designs (here the factors are gender and condition of the samples) was used for differential expression genes analysis. FDR P-value < 0.05 was used as the cut-off of significant differential expression genes. We

analysed the differential expressed gene sets corresponding to molecular pathways of the Reactome database [70].

## Statistical analysis

All graphs and statistical analyses were produced using GraphPad Prism 7 (GraphPad Software Inc., San Diego, CA, USA) as indicated in each figure legend. P values < 0.05 were deemed to be significant. In the graphs, values of male animals are displayed as triangles, while circles are used for females.

## Supporting information

**S1 Fig. Clinical scoring and PCA plots from transcriptomic analysis of tissues from SARS-CoV-2 experimentally infected hamsters.** (A) Hamsters were infected with Delta or BA.1 intranasally. Mock-infected animals received media alone. The weights and disease scores of each animal was recorded daily. (B) All animals were culled 2- or 6-days post-infection (dpi), and RNA was extracted from lungs and blood for RNAseq. PCA plots indicate the variance between the different samples at 2 and 6 dpi. Data were obtained from n = 8 animals per group from two independent experiments (4 females and 4 males per group).
(TIF)

**S2 Fig. Detection of viral protein or RNA in SARS-CoV-2 experimentally infected hamsters.** Micrographs of lung tissues collected from hamsters infected intranasally with either Delta or BA.1 (or mock-infected). Tissues were collected 2 days post-infection (2 dpi). Animals were culled at 2 dpi and lung sections were assessed for the presence of viral protein by immunohistochemistry, or viral RNA by *in situ* hybridisation.
(TIF)

**S3 Fig. Tissue Distribution of Delta and BA.1 in experimentally infected hamsters at day 1 post-infection.** (A) Hamsters were infected with either Delta or BA.1 intranasally (or mock-infected). Animals were culled 1 day post infection and turbinates, trachea and lungs were collected for digital pathology analyses. (B) Tissues were assessed for the presence of spike RNA by *in situ* hybridisation or (C) for the expression of MX1 by immunohistochemistry. For signal quantification, slides were scanned with an Aperio VERSA 8 Brightfield, Fluorescence & FISH Digital Pathology Scanner (Leica Biosystems) at 200 x brightfield magnification. (D) Nasal washes, throat swabs and lungs were analysed for the presence of SARS-CoV-2 genomic RNA by RT-qPCR. Statistical analysis was performed using an unpaired t test, *<0.05, **<0.01, ****<0.0001. Data is representative of two independent experiments, n = 6 (3 females and 3 males per group). Males: triangles; females: circles. Blue scale bar: 500 μm; green scale bar: 200 μm. Graphics made using biorender.com.
(TIF)

**S4 Fig. Lung histopathology of SARS-CoV-2 Delta-infected hamsters.** (A-B) Micrograph of lung sections stained with haematoxylin and eosin showing an infiltrate of neutrophils/heterophils (arrows, bars, 50 μm), and (C) lymphocytes (white arrow) and plasma cells (black arrow). The hyperplastic bronchial epithelium is shown with an asterisk (bar, 100 μm). (D) The diffuse distribution of lymphocytes is highlighted by immunohistochemistry by the presence of CD3[+] T cells throughout the lung (bar, 200 μm), and (E) plasma cells (PAX5[+]) around the vessels (asterisk) (bar, 100 μm). (F-H) Micrographs of serial sections of lung of a Delta-infected hamster assessed by immunohistochemistry. Infiltrating macrophages are highlighted using by IBA1staining. Note the presence of macrophages surrounding an area with severe proliferation of epithelial cells arranged in rosette-like structure, as established by E-cadherin (G) and TTF1

staining (H). The asterisk highlights the centre of the proliferation in F-Hs (ars = 400 μm.
(TIF)

**S5 Fig. Syncytia in the lungs of SARS-CoV-2-infected hamster.** (A) Micrographs of lung sections analysed by immunohistochemistry using TTF1 antibodies. Within the consolidated lung parenchyma, the presence of cells with multiple, partially fused nuclei can be detected in some animals (arrows and asterisk). These cells are identified as syncytia (black arrows). (B) Also the bronchial epithelium shows in some cases multinucleated cells interpreted as syncytia (black arrows) on the luminal side of the bronchus (highlighted with an asterisk). A and B, TTF-1 staining; bars, 50 micrometres.
(TIF)

**S6 Fig. Performance of the HALO trained algorithm to detect alveolar epithelial hyperplasia.** (A-B) Lung sections analysed by immunohistochemistry using TTF1 antibodies. TTF1 + cells include isolated type 2 pneumocytes and bronchial epithelial cells including artefacts of multilayered epithelium caused by sectioning (arrows). (C-D) Immunohistochemistry as in A-B analysed with HALO using the machine learning assisted module with the algorithm trained to ignore isolated type 2 pneumocytes and bronchial epithelium. Hyperplastic type 2 pneumocytes in rosette-like structures are highlighted in yellow by the software. Only these structures are quantified (shown in yellow). Asterisks highlight the lumen of bronchi. Bars, 200 micrometres.
(TIF)

**S7 Fig. Comparison of virulence in hamsters of SARS-CoV-2 variants used in this study.** Data shown in Fig 6A, 6D, 6E, 6G and 6H was merged for either (A) IBA1- or (B) TTF-1-positive areas (type 2 pneumocyte hyperplasia) by normalising results to those obtained in BA.1-infected hamsters (normalised to 1). Statistical analysis was performed using a One-Way ANOVA with Tukey's multiple comparisons test. Significance is indicated with *<0.05, **<0.01, ***<0.001, ****<0.0001. Male animals: triangles, female animals: circles.
(TIF)

**S8 Fig. Qualitative scoring of lung pathology of experimentally infected hamsters assessed by trained veterinary pathologists.** The amount of lung affected (A), respiratory epithelial cell or type II pneumocyte hyperplasia (B), alveolar inflammation (C), perivascular inflammation (D), bronchi/bronchioles inflammation (E) are shown as the average of the scores obtained in 6 animals per group. Each dot represents the value obtained by one Board-certified pathologist (not involved in the experimental phase of this study). The median is also highlighted. The final cumulative score (F) was calculated by adding per each animal the average scores by each pathologist (n = 6 hamsters per variant). Statistical analysis was performed using a One-Way ANOVA with Tukey's multiple comparisons test. Significance is indicated with *<0.05, **<0.01, ***<0.001, ****<0.0001.
(TIF)

**S9 Fig. Detection of type 2 pneumocyte hyperplasia in two distinct sections of the lungs of experimentally infected hamsters.** (A-B) Detection of alveolar epithelial hyperplasia (analysed as shown in S8 Fig) in hamsters experimentally infected with BA.1, BA.2 or BA.2.75. Data obtained in two distinct lung sections (Section 1 and 2) collected at approximately 100 μm from each other in the lung. Experiments have been processed, stained, scanned and analysed at different times.
(TIF)

## Acknowledgments

We acknowledge the assistance of Catrina Boyd, Scott McCall, and Nicola Munro at Biological Services at the University of Glasgow for guidance on animal experiments, and Lynn Oxford, Lynn Marion Stevenson, Frazer Bell, Jessica Lee, and Jan Duncan of the Veterinary Pathology Unit excellent technical assistance. We also gratefully acknowledge Diane Vaughan and Alana Hamilton of the III Flow Core Facility at the University of Glasgow for their support & assistance in this work.

## Author Contributions

**Conceptualization:** Gavin R. Meehan, Vanessa Herder, Arvind H. Patel, Massimo Palmarini.

**Data curation:** Gavin R. Meehan, Vanessa Herder, Natasha Palmalux.

**Formal analysis:** Gavin R. Meehan, Vanessa Herder, Jay Allan, Xinyi Huang, Georgios Ilia, Kyriaki Nomikou, Quan Gu, Sergi Molina Arias, Florian Hansmann, Alexandros Hardas, Charalampos Attipa, Nicole Upfold, Natasha Palmalux, Wilhelm Furnon.

**Funding acquisition:** Wendy S. Barclay, Arvind H. Patel, Massimo Palmarini.

**Investigation:** Gavin R. Meehan, Vanessa Herder, Jay Allan, Xinyi Huang, Karen Kerr, Diogo Correa Mendonca, Georgios Ilia, Kyriaki Nomikou, Nicole Upfold, Natasha Palmalux.

**Methodology:** Gavin R. Meehan, Vanessa Herder, Jay Allan, Georgios Ilia, Kyriaki Nomikou, Quan Gu, Natasha Palmalux, Jonathan C. Brown, Wilhelm Furnon.

**Project administration:** Gavin R. Meehan, Vanessa Herder, Arvind H. Patel, Massimo Palmarini.

**Resources:** Giuditta De Lorenzo, Vanessa Cowton, Jonathan C. Brown, Wendy S. Barclay, Wilhelm Furnon.

**Software:** Derek W. Wright, Quan Gu.

**Supervision:** Gavin R. Meehan, Vanessa Herder, Wendy S. Barclay, Ana Da Silva Filipe, Arvind H. Patel, Massimo Palmarini.

**Validation:** Gavin R. Meehan, Vanessa Herder.

**Visualization:** Gavin R. Meehan, Vanessa Herder, Massimo Palmarini.

**Writing – original draft:** Gavin R. Meehan, Vanessa Herder, Arvind H. Patel, Massimo Palmarini.

**Writing – review & editing:** Gavin R. Meehan, Vanessa Herder, Jay Allan, Xinyi Huang, Karen Kerr, Diogo Correa Mendonca, Georgios Ilia, Derek W. Wright, Quan Gu, Sergi Molina Arias, Florian Hansmann, Alexandros Hardas, Charalampos Attipa, Giuditta De Lorenzo, Vanessa Cowton, Nicole Upfold, Natasha Palmalux, Jonathan C. Brown, Wendy S. Barclay, Ana Da Silva Filipe, Wilhelm Furnon, Arvind H. Patel, Massimo Palmarini.

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
