## [Decision Letter · Decision Letter 0]

18 Sep 2023

Dear Prof. Palmarini,

Thank you very much for submitting your manuscript "Phenotyping the virulence of SARS-CoV-2 variants in hamsters by digital pathology and machine learning" for consideration at PLOS Pathogens. As with all papers reviewed by the journal, your manuscript was reviewed by members of the editorial board and by several independent reviewers. In light of the reviews (below this email), we would like to invite the resubmission of a significantly-revised version that takes into account the reviewers' comments.

The reviewers noted a number of major weaknesses in your manuscript describing a digital pathology pipeline for assessing differences between SARS-CoV-2 lineages. In your response, please pay particular attention to issues regarding sex as a biological variable, clarification of methodological details, the need for comparison of the pipeline with standard analysis by a certified pathologist and in a form that can be publicly shared.

We cannot make any decision about publication until we have seen the revised manuscript and your response to the reviewers' comments. Your revised manuscript is also likely to be sent to reviewers for further evaluation.

Sincerely,

Christiane E. Wobus

Academic Editor

PLOS Pathogens

Ana Fernandez-Sesma

Section Editor

PLOS Pathogens

Kasturi Haldar

Editor-in-Chief

PLOS Pathogens

orcid.org/0000-0001-5065-158X

Michael Malim

Editor-in-Chief

PLOS Pathogens

orcid.org/0000-0002-7699-2064

Reviewer's Responses to Questions

**Part I - Summary**

Reviewer #1: ‘Phenotyping the virulence of SARS-CoV-2 variants in hamsters by digital pathology and machine learning’ by Meehan et al presents a new method for determining COVID-19 variant severity based on quantification of alveolar epithelial hyperplasia in hamster lungs. They present IHC and ISH data for ISG expression, markers of virus replication, macrophage infiltration and also a machine trained, alveolar space specific analysis of TTF1 staining. The staining and quantification of these markers is robust, although the authors should be careful not to overstate differences that fail to reach statistical significance. Overall, the authors propose that TTF1 staining clusters can be used to reliably assess the pathogenesis of new COVID-19 variants and to quantitate moderate differences. While large differences in pathogenesis were readily detectable (ie Delta vs BA.1), TTF1 staining did not appear to be more sensitive than traditional WL measurements in differentiating Omicron substrains. Overall, while this method is intriguing and more biologically relevant than WL metrics, it is unclear that it has significant advantages to simple measurements of inflammatory cell infiltrate. Additionally, more information is needed about the robustness and sharability of the method.

Reviewer #2: This manuscript used histopathology and RNA-seq approaches to explore the pathology of SARS-CoV-2 variants of concerns (VOCs) in hamsters as an animal model. The authors also developed an online tool to visualize the differences upon infections in hamsters by different VOCs. The approaches appropriately addressed the research question. However, there are several minor points that will need authors to further clarify, to fulfill the accuracy of the study. With that, this manuscript requires minor revision.

Reviewer #3: In this work, the authors first compared the transcriptomic differences between Delta- and Omicron (BA.1)-infected hamsters. They then developed a semi-automatic and quantitative system using digital pathology and machine learning to assess histopathological damages in the COVID-19 Syrian hamster model. This was followed by further comparison of different Omicron sublineages.

Strengths:

1) The digital pathology pipeline has been used in assessing various diseases in human tissues and animal models, and has been proven to be gaining increasing acceptance among histopathologists. It allows less biased assessment of histopathological lesions and (semi)-quantitation.

2) The manuscript is generally well written. Except for a few clarifications needed, the majority of the methods and results are easy to follow.

Limitations:

1) While being useful, the use of digital pathology in infectious disease animal models is not too novel. The authors should consider further detailing the most novel points to increase the impact of the study.

2) There are some methodological issues that need further explanations and/or supplementary data as detailed below.

**Part II – Major Issues: Key Experiments Required for Acceptance**

Reviewer #1: Figure 4 - The authors should show either specific staining (preferred) or high resolution, highly magnified images to support their results text describing various neutrophil, lymphocyte and plasma cell infiltration and especially type 2 pneumocyte proliferation. These features cannot be observed from the H&E-stained sections that are shown.

The authors should use scientific language to describe their phenotypic observations not use the word medusa on it’s own to describe lung pathology. While medusa is an apt comparison, it is not on it’s own a meaningful descriptor.

Figure 6A – quantification of TTF1, IBA1 and MX1 shows no significant differences between BA.1 and BA.5 infected lungs while the text at line 236 says that the BA.5 lungs showed increased expression.

Was a specific lung orientation used for the cuts used in histopathological analysis? How much might different orientations impact the amount of airway vs alveolar space available to contribute to the alveolar epithelial hyperplasia that is measured since this is represented as percent positive area?

General/Line 303 – How was type II pneumocyte hyperplasia selected as the major histopathological phenotype to assess coronavirus pathogenesis? Was TTF1 staining compared to any other markers like Annexin-V for necrosis or others?

Many more details are needed about the AI training that was done select alveolar epithelial hyperplasia. “Several” slides were used to train the HALO algorithm, how many lungs from different infection conditions? Was this training confirmed for accuracy by a pathologist? Are there scripts that can be shared with other laboratories? Without a standardized, tested and sharable methodology there is no advantage to this process.

Reviewer #2: (No Response)

Reviewer #3: 1) It is problematic to mix male and female hamsters when the virulence of different virus strains is being assessed because it is well known that the sex and age of the hamsters may significantly affect the disease phenotype even when using the exact same virus strain. The authors must either show male and female as indicated by different symbols in each relevant figure, or they should supplement additional experimental data using the same sex (preferably male because the disease tends to be more severe in male hamsters and therefore the phenotypic differences would be easier to observe).

2) As a key part of this study is the development and validation of the digital pathology assessment pipeline, the same sample sections should be thoroughly assessed by at least 2 well-trained pathologists (who should be unaware of the digital pathology assessment’s findings and scores) as a “control gold standard” of the existing way to assess histopathological damage in the COVID-19 Syrian hamster model.

3) As correctly mentioned by the authors, a number of histopathology scoring systems for assessing SARS-CoV-2-induced respiratory tract tissue damage have been used in the field. Please add these scores either to the corresponding main figures or as supplementary figures.

**Part III – Minor Issues: Editorial and Data Presentation Modifications**

Reviewer #1: Figure 1 D and E – the gene names are too small to read

Figure S2 – what is the timepoint of these samples?

Figure S3 shows differences in spike RNA in turbinates and trachea only and no significant differences in Mx1 expression. Please modify the text at line 188 to reflect the data.

Line 182 – description of Figure 3 should mention the 2 DPI timepoint

Figure 3 – the lung staining images shown do not seem representative of the quantitation for Mx1 or IFIT1

Figure 4 – BA1 sections should show large airways for effective comparison with mock and Delta images.

Line 215 – please add at 6 dpi.

Figure 5 C – please explain the 3 panels, in particular what is the left most lung section?

Figure 5D – Are these panels in same samples/order as 5A and B?

Pax5 staining is mentioned in the materials and methods but does not seem to appear in the manuscript?

Line 262 refers to figure 6G, not 6H. 6H should be discussed in the text.

Line 310 – Did the authors demonstrate anywhere that infiltrating macrophages and proliferating epithelial cells make up most of the cells in the lung parenchyma? Neutrophils are an important and abundant cell type in the lung after coronavirus infection and should also be quantitated to allow for this statement. This would be an important supplemental figure to add to justify IBA1 and TTF1 staining as key markers of pathogenesis.

Line 150 vs Line 636 – one describes nucleocapsid IHC for figure S2 and one describes spike IHC, please clarify which is correct

Line 289 – The host immune response is also dictated by animal welfare and wellbeing prior to infection. These statements seem overly critical of WL and especially respiratory function as measures of disease.

Figure 6 – Please clarify in the text and figure legend that the TTF1 expression being quantitated is the software selected alveolar regions.

Reviewer #2: Line 385 in the cells and viruses. The authors clarified that the viruses were isolated from swab samples. The authors should then clarify in detail how the virus was purified to avoid any other viruses. For example, the authors may conduct PCR to detect other common respiratory viruses such as RSV, influenza, rhinovirus, etc. The authors may also do several rounds of serial dilution after virus passage before conducting PCR.

Line 576, legend for figure 4. I don't think the histopathology upon SARS-CoV-2 infection is equally distributed in the lung, unless there are papers describing it otherwise. So it is optimal to show the tissue from the same lung lobe at each timepoint.

Figures. The figures often use "positive area" as a parameter. However, in the methods section, it was not clear how the positive area was determined. In figure 3 for example, the background of D and E seems higher for both Delta and BA1, is it the reason their positive areas are bigger than A and B?

Reviewer #3: 1) Figures: The corresponding viral loads should be provided to allow easier correlation between histopathological damage, clinical scores, and viral burden in the figures.

2) Please make sure to spell out the full name of each terminology when it is used for the first time in the manuscript (eg: COVID-19 in line 45).

3) Lines 382-385: Please state explicitly whether the virus strains were plaque purified to avoid having mixed strains in the same sample.

PLOS authors have the option to publish the peer review history of their article (what does this mean?). If published, this will include your full peer review and any attached files.

Reviewer #1: No

Reviewer #2: **Yes: **Zhongyan Lu

Reviewer #3: No
---

## [Editor Report · Decision Letter 1]

30 Oct 2023

Dear Prof. Palmarini,

Thank you for your thoughtful and thorough revisions. We are pleased to inform you that your manuscript 'Phenotyping the virulence of SARS-CoV-2 variants in hamsters by digital pathology and machine learning' has been provisionally accepted for publication in PLOS Pathogens.

Best regards,

Christiane E. Wobus

Academic Editor

PLOS Pathogens

Ana Fernandez-Sesma

Section Editor

PLOS Pathogens

Kasturi Haldar

Editor-in-Chief

PLOS Pathogens

orcid.org/0000-0001-5065-158X

Michael Malim

Editor-in-Chief

PLOS Pathogens

orcid.org/0000-0002-7699-2064
---

## [Editor Report · Acceptance letter]

2 Nov 2023

Dear Prof. Palmarini,

We are delighted to inform you that your manuscript, "Phenotyping the virulence of SARS-CoV-2 variants in hamsters by digital pathology and machine learning," has been formally accepted for publication in PLOS Pathogens.

Best regards,

Kasturi Haldar

Editor-in-Chief

PLOS Pathogens

orcid.org/0000-0001-5065-158X

Michael Malim

Editor-in-Chief

PLOS Pathogens

orcid.org/0000-0002-7699-2064